# Automated BIM Reconstruction of Full-Scale Complex Tubular Engineering Structures Using Terrestrial Laser Scanning

**Jiepeng Liu** [1,2], **Lihua Fu** [1,2], **Guozhong Cheng** [1,2,*], **Dongsheng Li** [3], **Jing Zhou** [4], **Na Cui** [1,2] and **Y. Frank Chen** [5]

1    Key Laboratory of New Technology for Construction of Cities in Mountain Area, Chongqing University, Ministry of Education, Chongqing 400045, China; liujp@cqu.edu.cn (J.L.); fulihua@cqu.edu.cn (L.F.); nacui@cqu.edu.cn (N.C.)
2    School of Civil Engineering, Chongqing University, Chongqing 400045, China
3    College of Civil and Transportation Engineering, Shenzhen University, Shenzhen 518060, China; lds@cqu.edu.cn
4    China MCC5 Group Corporation Ltd., Chengdu 610000, China; 20165494@cqu.edu.cn
5    Department of Civil Engineering, The Pennsylvania State University, Middletown, PA 17057, USA; yxc2@psu.edu
\*    Correspondence: chengguozhong@cqu.edu.cn

**Abstract:** Due to the accumulation of manufacturing errors of components and construction errors, there are always deviations between an as-built complex tubular engineering structure (CTES) and its as-designed model. As terrestrial laser scanning (TLS) provides accurate point cloud data (PCD) for scanned objects, it can be used in the building information modeling (BIM) reconstruction of as-built CTESs for life cycle management. However, few studies have focused on the BIM reconstruction of a full-scale CTES from missing and noisy PCD. To this end, this study proposes an automated BIM reconstruction method based on the TLS for a full-scale CTES. In particular, a novel algorithm is proposed to extract the central axis of a tubular structure. An extended axis searching algorithm is applied to segment each component PCD. A slice-based method is used to estimate the geometric parameters of curved tubes. The proposed method is validated through a full-scale CTES, where the maximum error is 0.92 mm.

**Keywords:** complex tubular engineering structure; BIM reconstruction; terrestrial laser scanning; central axis-based modeling; central axis-based segmentation

## 1. Introduction

Complex tubular engineering structures (CTESs) consisting of straight and curved tubes have been widely used in stadiums, landmark towers, etc., because of their reasonable force distribution, high structural rigidity, aesthetic appearance, and good economic benefits [1,2]. The tubular components are connected to each other by cylindrical joints, presenting a uniform and attractive structure. However, due to the complex geometric shape of the CTES, the manufacturing and construction of tubular components are very difficult. The accumulation of the manufacturing errors of components and construction errors leads to deviations between an as-built CTES and its as-designed model. Therefore, to achieve life cycle management (LCM), such as construction quality assessment and component maintenance management [3–5], there is an increasing need for the building information modeling (BIM) reconstruction of as-built CTESs.

Recently, terrestrial laser scanning (TLS) technology with fast data acquisition capability and millimeter-level accuracy has been extensively used in BIM reconstruction applications in the architecture, engineering, and construction (AEC) industry [6–9]. TLS can be adopted to rapidly and accurately capture the point cloud data (PCD) of CTESs. Furthermore, BIM reconstruction can be performed based on the obtained PCD. Typically, the BIM reconstruction of as-built models based on PCD includes data segmentation and

object reconstruction. For the data segmentation, feature-based methods such as region growing [10] and random sample consensus (RANSAC) [11] are most commonly used. These methods allow for the segmentation of PCD based on the geometric features of scanned objects, which is applicable to CTESs with a single component type. However, due to the limited space available for the scanner to be set up at the construction site and the presence of occluding components, the obtained PCD may be missing and noisy (Figure 1), which will seriously affect the performance of feature-based methods. For object reconstruction, the geometric parameter estimation of the structure is essential to the existing BIM reconstruction studies of full-scale structures [12–14]. For a tubular component, the central axis and cross-sectional radius are the two critical geometric parameters, based on which an as-built model of a tubular component can be generated [15]. To estimate the central axis, it is necessary to first determine the points on the central axis and then connect these points to obtain the central axis [15–17]. To estimate the cross-sectional radius, the PCD of the tubular component is usually sliced onto the plane orthogonal to the central axis, to realize the estimation of the cross-sectional radius on the two-dimensional (2D) plane [15]. It is worth noting that the estimation of both the central axis and cross-sectional radius rely on local information from the PCD. The accuracy of the geometric parameter estimation can be significantly affected by the missing and noisy PCD of CTESs. Therefore, a BIM reconstruction method that can extract the accurate geometric parameters from the missing and noisy PCD of tubular components is urgently required. Moreover, few studies have conducted the BIM reconstruction of a full-scale CTES. The segmentation of straight tubes was conducted in [16,17], which is not suitable for segmenting a full-scale CTES with curved tubes. These normal-based region growing methods [15] are suitable for segmenting a full-scale CTES with curved tubes. However, computation of normal vectors is subject to noisy or missing data. Therefore, this study attempts to fill these voids.

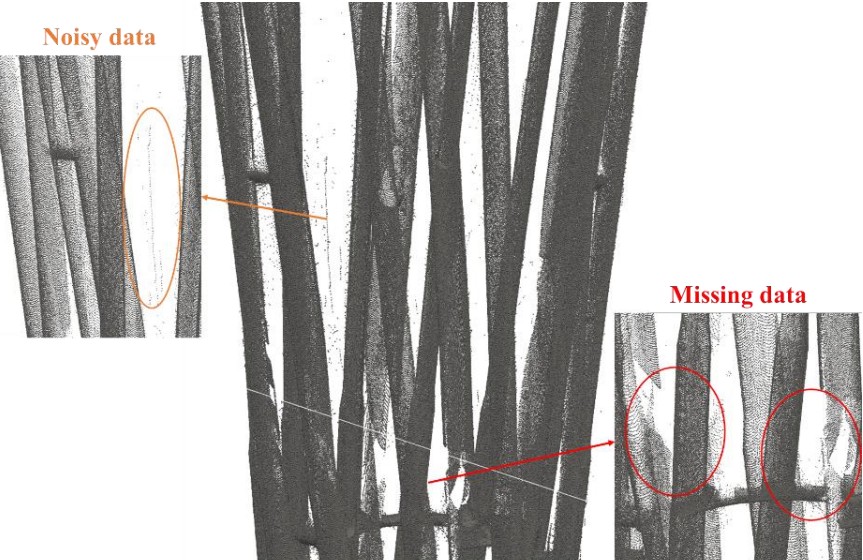

**Figure 1.** The obtained PCD with missing and noisy data.

To address the above problems, this study proposes an automated BIM reconstruction method to generate an as-built model for a full-scale CTES based on the TLS. The main contributions of this study include: (1) A novel central axis extraction method for tubular components is developed; (2) an extended axis searching algorithm based on the concept of region growing is proposed to segment PCD of CTES with missing and noisy data.

The reminder of this paper is organized as follows. Section 2 describes the work related to the PCD segmentation and BIM reconstruction of tubular components. Section 3 gives the details of the proposed BIM reconstruction method. Section 4 presents details of the validation experiment. Finally, Section 5 summarizes and concludes this study.

## 2. Related Work

The procedure of the BIM reconstruction from the scanned PCD usually includes data segmentation and object reconstruction. For the data segmentation, software-aided and semi-automatic techniques have dominated the PCD segmentation. Various standalone computer programs (e.g., Trimble RealWork [18], Geomagic Wrap [19], and Edgewise [20]) allow users to manually segment PCD. However, the manual segmentation of PCD is time-consuming and labor-intensive for CTES. Thus, there is a need to develop a more effective automated PCD segmentation algorithm. Several algorithms such as random sample consensus (RANSAC) [11], Hough transform [21], and region growing [22] have been widely adopted in the automated PCD segmentation and object reconstruction of tubular components. For example, Schnabel et al. [11] adopted the RANSAC algorithm to segment the PCD of tubular components. This method overcomes the effects of noise by iteratively estimating the best model. However, the method [11] relying on the parameters calculated from the normal of the component surface is subject to missing data. Liu et al. [23] proposed a circle fitting method to detect the tubular component in the projected ground plane, which is not suitable for curved tubular components with a complex interlocking relationship. Moreover, Kazuaki et al. [15] proposed a region growing algorithm based on the normal of a component surface to extract the PCD of tubular components. However, the PCD segmentation for a CTES based on the normal and curvature [3–5] is very sensitive to noise. In addition, the missing data caused by the limited space available for the scanner to be set up at the construction site and the presence of occluding components can lead to errors in the estimation of normal and curvature, resulting in inaccurate segmentation results. To avoid directly calculating the normal and curvature of the tubular components for the PCD segmentation, Lee et al. [17] proposed extracting the central skeleton of the tubular components and segmenting the PCD based on the central skeleton nodes, but the extracted central skeleton will be shifted in the missing data, which cause errors in skeleton nodes. Hence, more reliable algorithms for automated data segmentation based on the central axis are needed.

For the object reconstruction, since the radius of the curved tubular components of CTES will change with the length depending on the design requirements [24], the following two critical geometric parameters are required: central axis and the cross-sectional radius. As mentioned above, the RANSAC algorithm [11] can determine the linear vector direction of the central axis of a tubular component by using the normal of the component surface, but the missing data can affect the estimated normal and lead to an inaccurate central axis. The classical Hough transform has been extended to the extraction of three-dimensional (3D) geometric objects like cylinders [25]. The method of determining a geometric parameter by iteratively fitting to update the parameter takes too much time. Patil et al. [26] proposed an improved Hough transform method for estimating the geometric parameters from a tubular component, which suggests that using the region-based adaptive method will continuously limit the search area to reduce the number of iterations. However, the execution time increases significantly as the PCD grows, in which is unsuitable for the rapid model generation of large-scale PCD. To reduce the error of a normal estimation from the missing data when extracting the geometric parameter, Guo et al. [27] developed the fast minimum covariance determinant (FMCD) method to reduce the influence of outliers when calculating the principal components, while it has a huge computational burden. The data simplification can reduce the execution time, but the sampled missing data will result in more information loss, which may cause a worse impact on the parameter estimation. Kazuaki et al. [15] proposed slicing the tubular component data and projecting the data onto a plane orthogonal to the central axis to perform cross-sectional radius estimation. This method is effective, but the precision depends on the accuracy of the central axis estimation. Jin et al. [16] proposed a rolling sphere algorithm to extract the central axis of tubes, which obtains a central axis with a long execution time. The central axis obtained by this method is noisy due to the missing data. Apart from this, a method based on the Laplacian smoothing to extract the central skeleton [28,29] has recently been proposed,

which has been used to quickly obtain the central axis of a tubular component and is robust to the sampling of data [17]. However, the central skeleton extracted by the Laplacian contraction algorithm will deviate to a certain extent when there is missing data. Therefore, considering the issues mentioned above, it is desirable to propose an algorithm that can quickly and accurately extract the central axis of a tubular component from the PCD with missing and noisy data.

## 3. Methodology

In this study, an automated BIM reconstruction method is proposed to generate the as-built model from the PCD of a CTES, which involves four steps: central axis extraction, PCD segmentation, geometric parameter estimation, and model generation. The details of each step are described in Section 3.1, Section 3.2, Section 3.3, Section 3.4, respectively. The flowchart of the proposed method is shown in Figure 2.

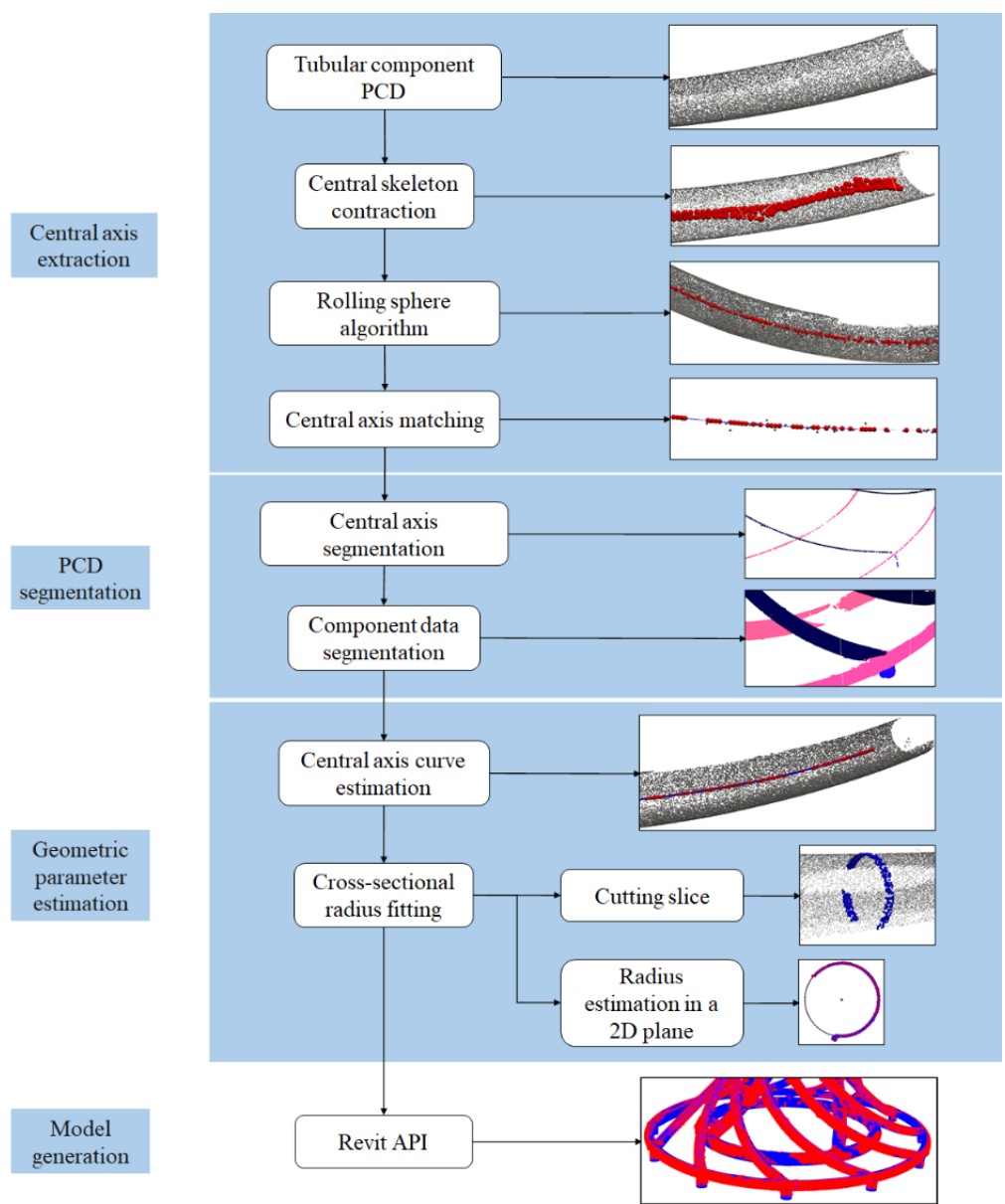

**Figure 2.** The flowchart of the proposed method.

### 3.1. Central Axis Extraction

As described in Section 1, the central axis is one of the critical geometric parameters for tubular components. The tubular components of the CTES in this study are so close to each other that the scanned data have more missing data due to structural occlusions. Currently, the central skeleton contraction algorithm [16] and rolling sphere algorithm [17] are widely adopted to extract the central axis. However, the central skeleton contraction algorithm is susceptible to missing and noisy data, and the computation cost of the rolling sphere algorithm is very high for large-scale PCD. Therefore, a novel central axis extraction method is developed by integrating the central skeleton contraction algorithm and rolling sphere algorithm. Subsequently, a central axis refinement algorithm is developed. As shown in Figure 2, the central skeleton contraction algorithm is first used to calculate the initial candidates on the central axis. Then, taking these candidates as the center of a sphere, the sliced PCD of the tubular component is selected and used to determine better candidates by using the rolling sphere algorithm. Finally, the central axis refinement algorithm is adopted to extract the accurate central axis.

#### 3.1.1. Central Skeleton Contraction Algorithm

The skeleton of a 3D object is an abstract representation of the geometric and topological set of 3D shapes [30,31]. This study uses the method proposed by Cao et al. [28] to extract the skeleton of the PCD of tubular components, including the one-ring neighborhood construction, Laplacian matrix generation, and Laplacian contraction. The points belonging to the skeleton are regarded as the initial candidates on the central axis.

*One-ring neighborhood construction*: To ensure the uniformity of the neighborhood sampling, the neighborhood of each point is constructed by the k-nearest neighbors (kNN) algorithm [32] with $k = 0.012 \times samples$ [28], where "*samples*" denotes the total number of points. Denoting $N_k(p_i)$ as the kNN of a point $p_i$, the principal component analysis (PCA) algorithm [33] can be used for dimensional reduction, as shown in Figure 3a. $N_k(p_i)$ can be projected onto the plane vertical to the principal component. Delaunay triangular dissection [34] is then performed in the projection plane. Figure 3b shows the projected points $Proj(p_{i,1}, p_{i,2}, \ldots, p_{i,k})$ in the Delaunay triangular dissection that constitute the one-ring neighborhood of the projected point $Proj(p_i)$. Note that, as the iteration process compresses, the neighborhood area gradually becomes smaller and makes the calculation of the tangent plane difficult. Therefore, during the iteration, the corresponding points in the original PCD of $N_k(p_i)$ should be found and the one-ring neighborhood is constructed on the tangent plane that is calculated based on these points.

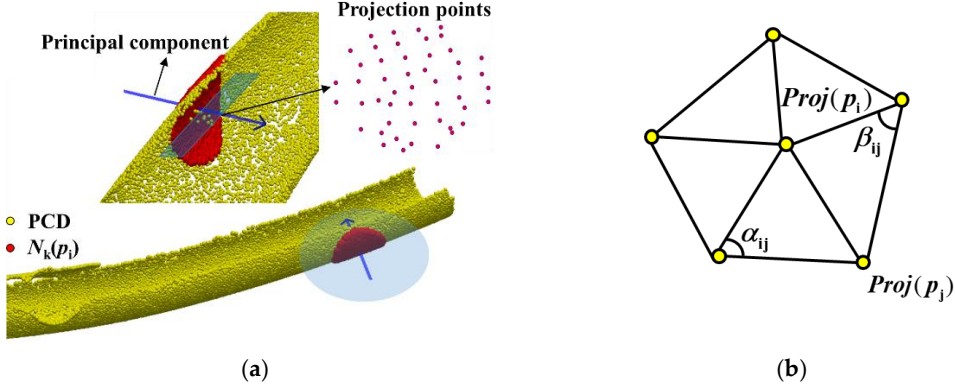

(a)  (b)

**Figure 3.** The Delaunay triangular dissection of the PCD neighborhood: (**a**) the construction of the one-ring neighborhood; (**b**) the example of the Delaunay triangular dissection.

*Laplacian matrix generation*: This study adopts the cotangent weighting scheme to generate the Laplacian matrix from the constructed one-ring neighborhood. The scheme

for calculating the cotangent weights of the Laplacian matrix presented in [35] is given as follows:

$$L_{ij} = \begin{cases} cot\alpha_{ij} + cot\beta_{ij} & if\ i \neq j\ and\ (i,j) \in \Gamma_i \\ \sum_{o=1}^{k} -w_{io} & if\ i = j \\ 0 & others \end{cases}, \tag{1}$$

where $\alpha_{ij}$ and $\beta_{ij}$ are the opposite angles of the two triangles shown in Figure 3b, $\Gamma_i$ is a planar Delaunay triangulation of $Proj(p_{i,1}, p_{i,2}, \dots, p_{i,k})$, and $E$ is a triangular face piece in the triangular dissection.

*Laplacian contraction*: To obtain the skeleton of the PCD of tubular components $P'$, the following quadratic energy function needs to be solved:

$$\min_{P'} \| W_L L P' \|^2 + \sum_i W_{H,i}^2 \| p_i' - p_i \|, \tag{2}$$

where $L$ is a Laplacian square matrix of $n \times n$ with cotangent weights; $W_L$ and $W_H$ are the $n \times n$ diagonal matrices controlling the forces of contracting and maintaining the original position, respectively; and $W_{L,i}$ and $W_{H,i}$ are the $i$th diagonal element of $W_L$ and $W_H$. Solving Equation (2) is equivalent to iteratively solving the following linear system [29]:

$$\begin{bmatrix} W_L \\ W_H \end{bmatrix} P' = \begin{bmatrix} 0 \\ W_H P \end{bmatrix}. \tag{3}$$

During the iterative process, $W_L$ is iteratively updated with $W_H$ by the following equations:

$$\begin{cases} W_L^{t+1} = s_L W_L^t \\ W_{H,i}^{t+1} = W_{H,i}^t \times \sqrt{\dfrac{S_i^t}{S^0}} \end{cases}, \tag{4}$$

where $s_L$ is the update operator of contraction weights, $S_i^t$ is the current one-ring neighborhood area of $Proj(p_i)$, and $S^0$ is the mean of the initial one-ring neighborhood area of $Proj(p_i)$. The iteration terminates when $(S_i^{t+1} - S_i^t)/S^0$ is less than a preset threshold. The parameters are set as suggested by Cao et al. [28]. Figure 4 shows the extracted central skeleton of the PCD of a tubular component after four iterations, where the extracted central skeleton is affected by the missing data.

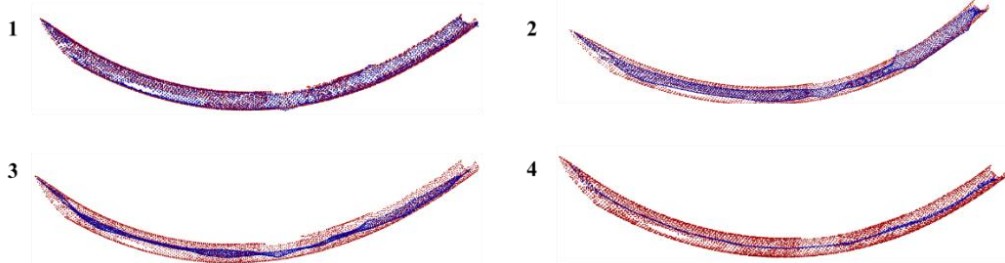

**Figure 4.** The performance of skeleton contraction technique performance for a curved tubular component with missing data.

3.1.2. Rolling Sphere Algorithm

Since the central skeleton extracted by the Laplacian contraction algorithm is deviated in the case of missing PCD of tubular components (Figure 4), the rolling sphere algorithm is adopted in this study to improve the accuracy of the candidates on the central axis. As shown in Figure 5, applying the rolling sphere algorithm to estimate the central axis is based on the centroid trace left by a sphere with the same radius as the tube rolling inside the tubular component. A rolling sphere model can be defined as a mathematical model of a sphere with a line, which is given as follows:

$$(x - x_0)^2 + (y - y_0)^2 + (z - z_0)^2 - r^2 = 0 \tag{5}$$

$$\frac{x - x_0}{a} = \frac{y - y_0}{b} = \frac{z - z_0}{c} = k,\tag{6}$$

where $(x_0, y_0, z_0)$ and $r$ are the center point and radius of the sphere, respectively; and $(a, b, c)$ is the linear direction of the center point.

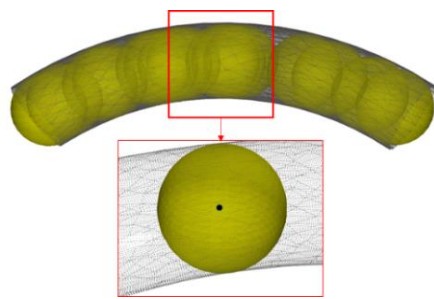

**Figure 5.** Central axis estimation using the trace of rolling sphere.

To determine the above parameters, the principal axis direction of each point on the central skeleton can be first calculated using the PCA algorithm based on its neighboring points. This principal axis direction is the rough linear direction $(a, b, c)$ of the central point. Then, the PCD of the tubular component can be sliced along $(a, b, c)$. Finally, $(x_0, y_0, z_0)$ and $r$ can be estimated from the sliced data using the RANSAC algorithm, as shown in Figure 6, in which the central skeleton is in blue, the PCD of the tubular component are in gray, and the sliced data are in red. Since the minimum interval between two neighboring tubular components is 1.5 R, $r$ should be less than 1.5 R in this study. Moreover, the slice length was set to 0.15 R to achieve a balance between computation time and accuracy, as illustrated in Figure 7.

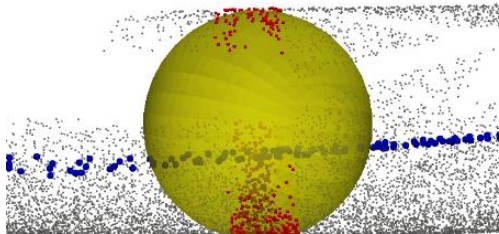

**Figure 6.** An example of the proposed method for fitting the sphere from missing data.

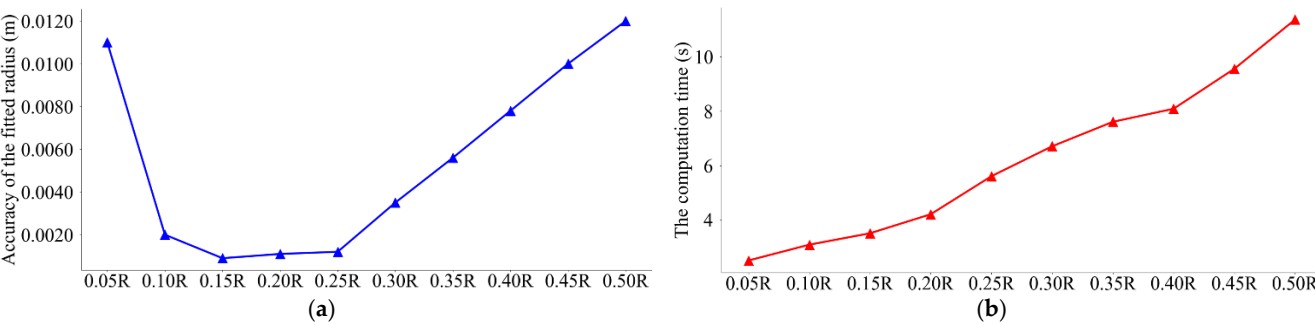

**Figure 7.** Algorithm parameter analysis of the length of the slice: (**a**) accuracy of fitted radius with different values of the parameter; (**b**) the computation time with different values of the parameter.

### 3.1.3. Central Axis Refinement

Although fitting the sphere using the RANSAC with sliced data can overcome the effects of missing data and noise, the obtained spherical centers will still show dispersion. To obtain an accurate central axis, a central axis refinement algorithm is developed to optimize the candidates obtained in the last step.

The aim of the central axis refinement is to update the position of candidates that are consistent in the linear direction. Hence, the linear direction of each candidate is first calculated through its kNN, as shown in Figure 8a. Then, the neighboring points with the same linear direction as the calculated candidate are found in the 2*k* neighborhoods of each candidate. The position of the new candidate is the average of these points that has the same direction as the calculated candidate, as shown in Figure 8b. The central axis refinement result of a tubular component is shown in Figure 8c, and the pseudocode of the central axis refinement algorithm is given in Algorithm 1, where the angle threshold $\alpha$ is set to 20° according to Jin et al. [16].

---

**Algorithm 1:** The pseudocode of the central axis refinement algorithm.

| | |
|---|---|
| **input:** | central axis candidates $C = \{c_i | i = 1, \ldots, n\}$; angle threshold $\alpha$; $k$ |
| **output:** | refined central axis $\Omega$ |

1      initialize refined central axis set $\Omega = \varnothing$
2      compute the neighborhood candidates $\{N_k(c_i)\}$ and $\{N_{2k}(c_i)\}$
3      compute the linear direction vectors $\{v_i\}$ based on $\{N_k(c_i))\}$
4      **for** $i = 1, \ldots, n$
5          initialize $C_m = \varnothing$
6          **for** $j = 1, \ldots, 2k$
7            compute the vector angle $\theta_{ij}$ between $v_i$ and $v_{i,j}$
8            **if** $\theta_{ij} \leq \alpha$
9              $C_m = C_m \cup \left\{ c_{i,j} \right\}$
10          **end if**
11          **end for**
12          **if** $card(C_m) > 1$
13            compute the average $s_i$ of $C_m$
14            $\Omega = \Omega \cup \{s_i\}$
15          **end if**
16      **end for**
17      **return** $\Omega$

---

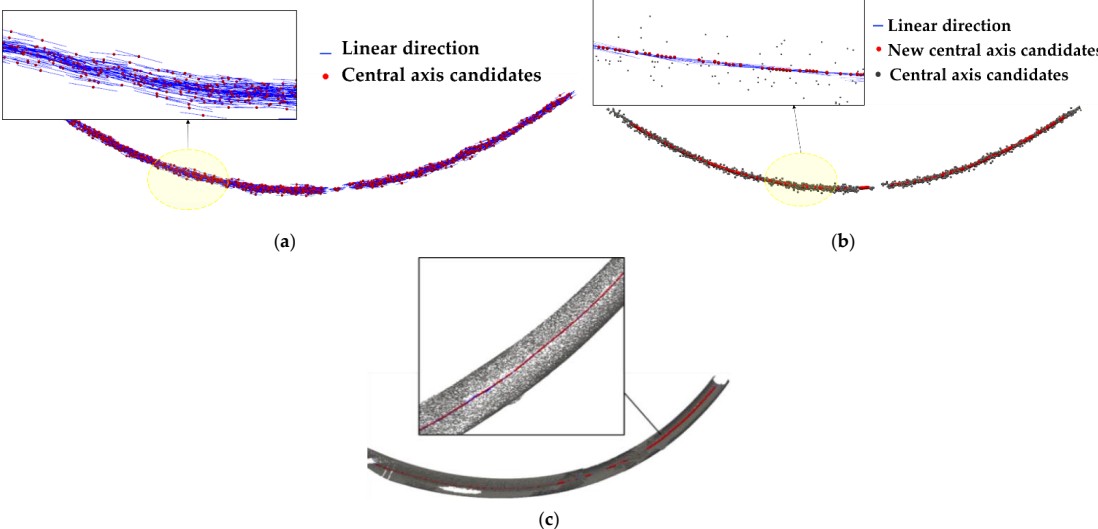

**Figure 8.** Central axis refinement effects: (**a**) linear direction vectors of central axis candidates; (**b**) obtained new candidates; (**c**) the central axis refinement result of a tubular component.

### 3.2. PCD Segmentation

Since feature-based segmentation algorithms are time-consuming and have poor robustness to noisy and missing data, this study proposes to segment the PCD of each tubular component based on the obtained central axis, which completely records the overall

shape of the entire CTES. Therefore, the PCD segmentation of CTES in this study consists of the following two steps: central axis segmentation and component data segmentation.

### 3.2.1. Central Axis Segmentation

There are two main types of central axis of curved tubes, which are circular and noncircular. It is worth noting that the central axis with circular shape can be easily segmented using the RANSAC algorithm to extract the sphere from the points on the central axis. Therefore, this section focuses on the extraction of noncircular central axis.

In this study, an extended axis searching algorithm based on the concept of region growing [22] is proposed for the segmentation of the noncircular central axis. The algorithm starts from a random point $s_i$ (green), shown in Figure 9a. Its neighboring points $N_k(s_i)$ iteratively expand from both ends within a certain radius, as shown in Figure 9b,c, in which $N_k(s_i)$ is in red and the expanded points are in yellow and blue at different ends. A point is added when its linear direction coincides with the linear direction of the endpoint, as shown in Figure 10. The angle $\theta_{ij}$ between two linear direction vectors $v_i$ and $v_j$ is calculated as follows:

$$\theta_{ij} = arccos\left(\frac{v_i \times v_j}{\parallel v_i \parallel \parallel v_j \parallel}\right). \tag{7}$$

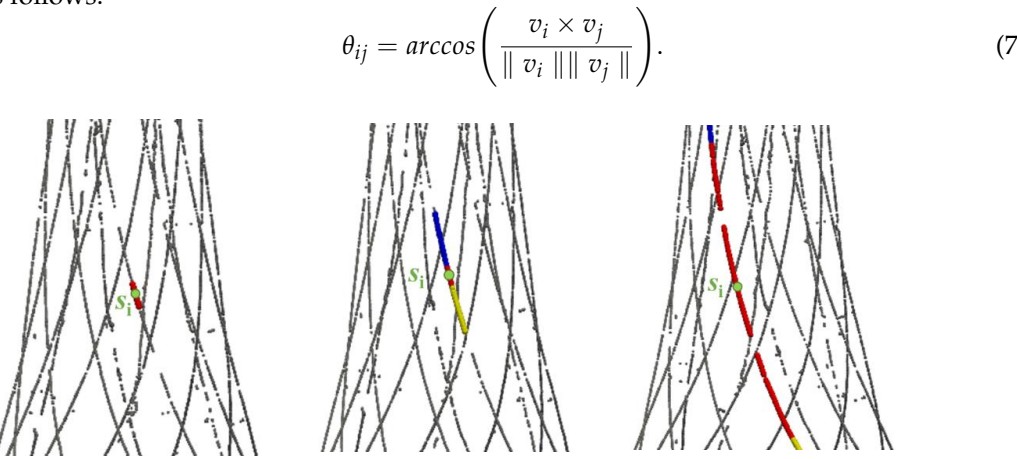

| (a) | (b) | (c) |

**Figure 9.** Extended axis searching algorithm for segmenting central axis: (**a**) initial state; (**b**) neighborhood extension; (**c**) the result of the seventh iteration.

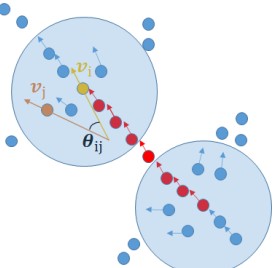

**Figure 10.** The calculated angle $\theta_{ij}$ between $v_i$ and $v_j$.

It is worth noting that the search radius needs to be large enough since the central axis is usually discontinuous due to missing data. The points on the central axis that are in a different linear direction from the endpoints can be filtered out by an angle threshold. The iteration terminates when the number of points of the segmented central axis stops changing. Algorithm 2 presents the pseudocode for the extended axis searching algorithm. In this study, the angle threshold is set to 20° and the search radius is set to 1.5$R$ empirically.

| **Algorithm 2:** The pseudocode for the extended axis searching algorithm. | |
|---|---|
| **input:** | central axis $S = \{s_i \mid i = 1, \ldots, n\}$, angle threshold criterion $\theta$, the search radius $r_s$ |
| **output:** | segmented central axis $S_g$ |
| 1 | initialize segmented central axis set $S_g = \varnothing$ |
| 2 | compute the neighborhood points $\{N_k(s_i)\}$ of each point on the central axis |
| 3 | compute the linear direction vectors based on $\{N_k(s_i)\}$ |
| 4 | $c = 1$ |
| 5 | **while** $S \neq \varnothing$ |
| 6 |     select a random point $s_i$ and $N_k(s_i)$ |
| 7 |     remove $s_i$ from $S$ |
| 8 |     initialize $R_c = N_k(s_i)$, *change_point* $= 1$, $t = 1$ |
| 9 |     **while** *change_point* $> 0$ |
| 10 |         extract two ends $s_{i1}$, $s_{i2}$ of $R_c$ |
| 11 |         compute the extended points $E(s_{i1})$, $E(s_{i2})$ based on $r_s$ |
| 12 |         compute the angle from each point in $E(s_{i1})$ to $s_{i1}$ and from each point in $E(s_{i2})$ to $s_{i2}$ |
| 13 |         find the points $R_{ct}$ on the central axis that their angles are less than $\theta$ |
| 14 |         upgrade $R_c = R_c \cup R_{ct}$, *change_points* $= card(R_{ct})$, $t = t + 1$ |
| 15 |     **end while** |
| 16 |     **if** $card(\psi_c) > k$ |
| 17 |         record the category of current segmented central axis $R_c$ as c |
| 18 |         add the labelled $R_c$ to $S_g$ |
| 19 |         remove $R_c$ from S |
| 20 |         $c = c + 1$ |
| 21 |     **end if** |
| 22 | **end while** |
| 23 | **Return** $S_g$ |

### 3.2.2. Component Data Segmentation

After realizing the central axis segmentation, the PCD of each tubular component can be segmented by searching the neighborhood in the overall PCD with the points on the central axis according to a certain search radius. However, there are some noisy data in the overall PCD. To improve the accuracy of the geometric parameter estimation of tubular components, the noisy data need to be filtered while performing a data segmentation.

As shown in Figure 11, the PCD of a tubular component *P* can be selected according to the following equation:

$$P = \{p_i \mid (d_i < \omega \times r) \& |\eta_i| < \varepsilon\} \tag{8}$$

$$d_i = |p_i - s_i| - r, \tag{9}$$

where *r* is the radius of the fitted sphere whose center $s_i$ is on the central axis, $d_i$ is the distance from $p_i$ to the fitted sphere calculated by Equation (9), and $\eta_i$ is the angle between the normal vector $n_i$ and the unit vector $\overrightarrow{s_i p_i}$. In Equation (8), the first term is the distance control condition, and the second term is the angle control condition. In this study, $\omega$ is set to 0.05 and $\varepsilon$ is set to 0.1 according to [36].

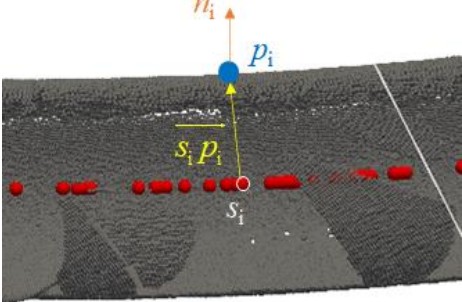

**Figure 11.** An illustration of the component data segmentation.

### 3.3. Geometric Parameter Estimation

To obtain a solid model of a tubular component, the curve of the central axis needs to be obtained and the cross-sectional radius needs to be estimated.

#### 3.3.1. Central Axis Curve Estimation

In this study, the Catmull–Rom algorithm [37] is adopted to estimate the curve of the central axis. Catmull–Rom splines give the tangent $\tau(p_{i+1} - p_{i-1})$ at each point $p_i$ with the previous point $p_{i-1}$ and next point $p_{i+1}$. Consider a single curve segment $p(u)$ between $p_{i-1}$ and $p_i$. In a 3D space, it can be expressed by a cubic function $p(u) = \sum_{k=0}^{3} c_k u^k$, where $u$ is an independent variable that is the ratio of the distance from a point on $p(u)$ to $p_{i-1}$ to the distance from $p_i$ to $p_{i-1}$; and $c_k$ is a coordinate point in the 3D space. The slope of $p_{i-1}$ is determined by $p_{i-2}$, $p_i$ and the slope of $p_i$ is determined by $p_{i-1}$, $p_{i+1}$. Therefore, $p(u)$ can be defined by four points, $p_{i-2}$, $p_{i-1}$, $p_i$, and $p_{i+1}$; and $c_k$ can be calculated from a linear combination of these four points. Therefore, $p(u)$ is obtained from a linear combination of the four points by the following equation:

$$p(u) = u^T M p = \begin{bmatrix} 1 & u & u^2 & u^3 \end{bmatrix} \begin{bmatrix} 0 & 1 & 0 & 0 \\ -\tau & 0 & \tau & 0 \\ 2\tau & \tau-3 & 3-2\tau & -\tau \\ -\tau & 2-\tau & \tau-2 & \tau \end{bmatrix} \begin{bmatrix} p_{i-2} \\ p_{i-1} \\ p_i \\ p_{i+1} \end{bmatrix}, \tag{10}$$

where the parameter $\tau$ is known as "tension," which affects how sharply the curve bends at the four points. According to [37], $\tau$ is set to 0.2 in this study.

Before applying the Catmull–Rom algorithm, the order of the points on the central axis needs to be determined for the curve configuration. To determine the order of the points on each segmented central axis, the points can be arranged along the direction of the first principal axis. Then, the Catmull–Rom algorithm can be used for the curve estimation. Figure 12 shows the estimated central axis curve (blue) of the central axis points (red) of a tubular component.

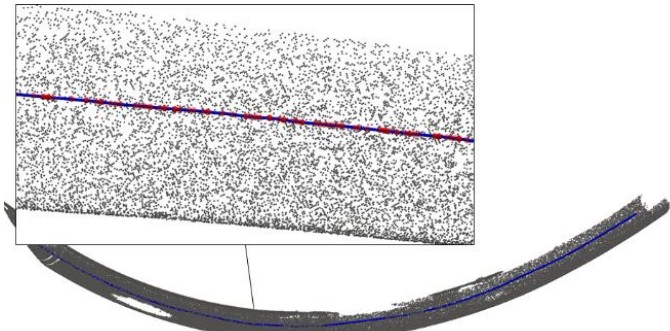

**Figure 12.** An example of the central axis curve estimation.

#### 3.3.2. Cross-Sectional Radius Fitting

To accommodate the tubular components with variable cross sections, a slice-based method is adopted to estimate the cross-sectional radius, as shown in Figure 13. The PCD is cut along the central axis curve and the slice is perpendicular to the central axis curve, as shown in Figure 13a. Figure 13b shows that the cross-sectional radius is estimated in a two-dimensional (2D) plane using the PCA and RANSAC algorithms. Finally, the model of the tubular component generated from the estimated parameters is shown in yellow in Figure 13c.

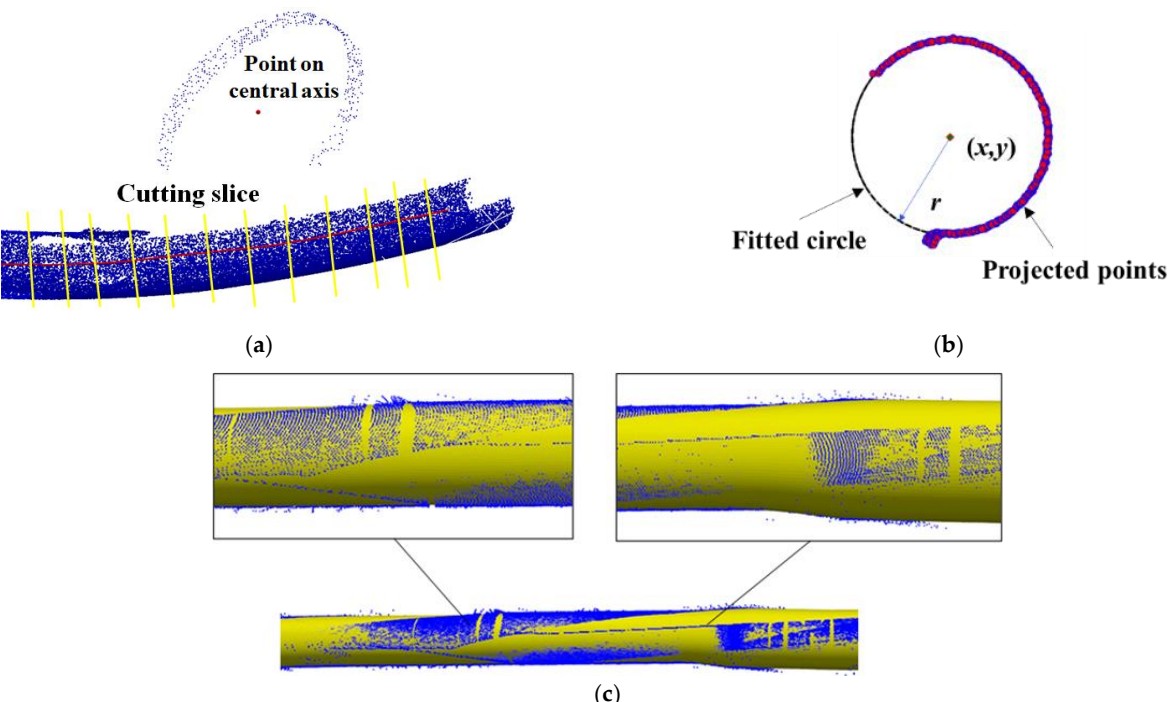

**Figure 13.** An example of cross-sectional radius fitting: (**a**) cutting slice; (**b**) the estimation of a cross-sectional radius; (**c**) the modeling result of a tubular component with variable cross sections.

### 3.4. Model Generation

The results of the geometric parameter estimation of tubular components, including the cross-sectional radius $r$, linear direction vector $v$, and centers $s$ of slices with 5 cm, are sorted in a text file. A solid-based family is adopted to create tubular components through the Revit application programming interface (API) [38]. The coordinates of two ends of a slice can be calculated based on $s$ and $v$. The command "sweep" in the API is used to create a straight cylinder with both ends as the origin and terminal points respectively. The as-built model of the CTES can be automatically generated from these straight cylinders, and can then be applied to the subsequent LCM.

## 4. Validation Experiment

To validate the proposed BIM reconstruction method for as-built CTES, a real-world CTES was scanned using FARO S150 [39]. A total of five scans were captured at different locations. The scan resolution is 20,480 laser scan points per scanning profile. The PCDs were registered using sphere targets, and the average error of the PCD registration was 0.31 mm, within an acceptable range. The registered PCD of the CTES includes 22,213,417 points. Section 4.1 presents the comparison test of the proposed central axis extraction method with the rolling sphere algorithm. Then, Section 4.2 shows the experimental results of the BIM reconstruction from the actual PCD of the CTES. The accuracy of the reconstructed BIM is presented in Section 4.3. The application of construction quality assessment based on the as-built model is given in Section 4.4. Section 4.5 shows an experiment of BIM reconstruction of a similar CTES using the proposed method.

### 4.1. Comparison Test

A set of sampled missing and noisy PCD of a tubular component including 1157 points was used in this test, as shown in Figure 14a. Table 1 lists the evaluation indicators of the proposed central-axis extraction method and rolling sphere algorithm, including the running time and average and variance of the radius for fitted spheres. The extraction results for the candidates on the central axis by the two methods are shown in Figure 14b.

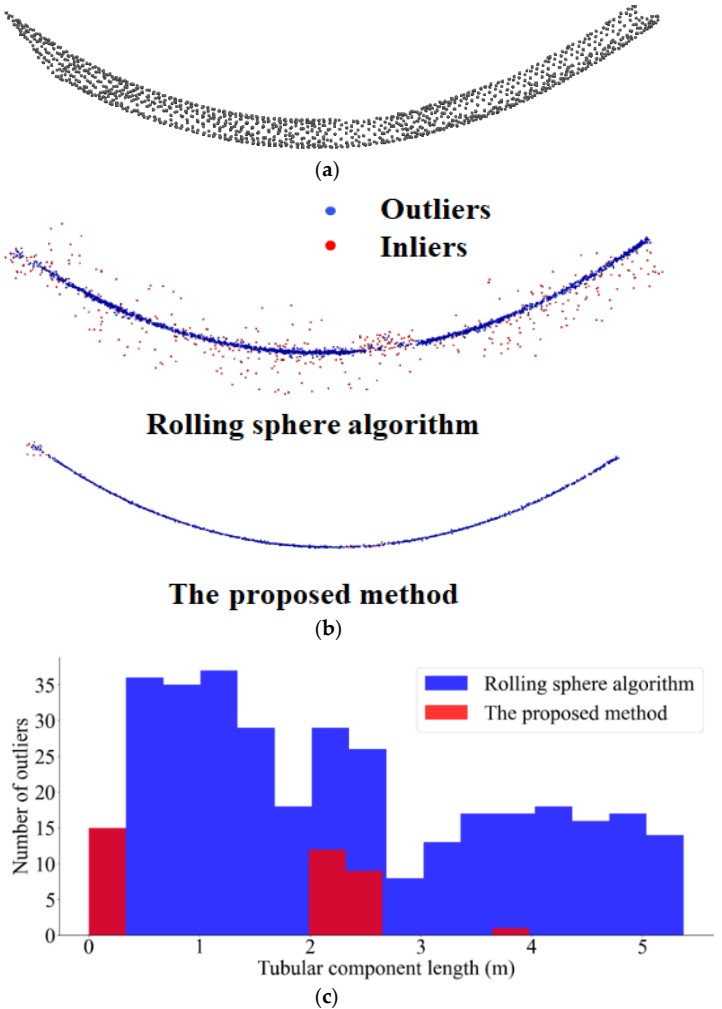

**Figure 14.** Extraction results for the candidates on the central axis: (**a**) sampled missing and noisy PCD of a tubular component; (**b**) the candidates obtained by the two methods; (**c**) the statistical histogram of the outliers from the candidates obtained by the two methods.

**Table 1.** The evaluation indicators for the proposed method and the rolling sphere algorithm.

| Evaluation Indicators | Rolling Sphere Algorithm | The Proposed Method |
| --- | --- | --- |
| Execution time (s) | 208 | 165.47 |
| Average of the radius for fitted spheres (m) | 0.190 | 0.176 |
| Variance of the radius for fitted spheres | 0.073 | 0.0034 |

As indicated in Table 1, the execution time of the rolling sphere algorithm is about 1.3 times that of the proposed method. This is because the rolling sphere algorithm is required to fit the sphere twice for each point to obtain the candidates. As shown in Figure 14b, the extraction results of both methods are processed using a neighborhood-based statistical filtering method [40]. The radius of the neighborhood is set to 0.1 m and the threshold of the number of neighboring points is set to 10. If the number of neighboring points is greater than the threshold, it is defined as an inlier, and as an outlier if vice versa. The comparison of the outliers filtered from the candidates obtained by the two methods is shown in Figure 14c. According to Figure 14b,c and the average and variance of the radiuses for fitted spheres given in Table 1, the rolling sphere algorithm provides more discrete results for the calculated candidates, suggesting a need for more smoothing of the candidate points. As the ground truth of the cross-sectional radius of this tested tubular

component is 0.175 m, the proposed method shows better efficiency and effectiveness for missing and noisy data compared to the rolling sphere algorithm.

### 4.2. Experimental Results for the BIM Reconstruction

The input PCD of the CTES includes 22,213,417 points, as shown in Figure 15a. The PCD is sampled using the voxel grid method [41], with the size being 0.1 m for rapid central skeleton contraction. The proposed method is performed on a personal computer with i7-7700k CPU @ 4.20 GHz. The execution time and number of points of each step are given in Table 2. Figure 15 shows the processing results of each step and Figure 16 shows the reconstructed as-built model in BIM.

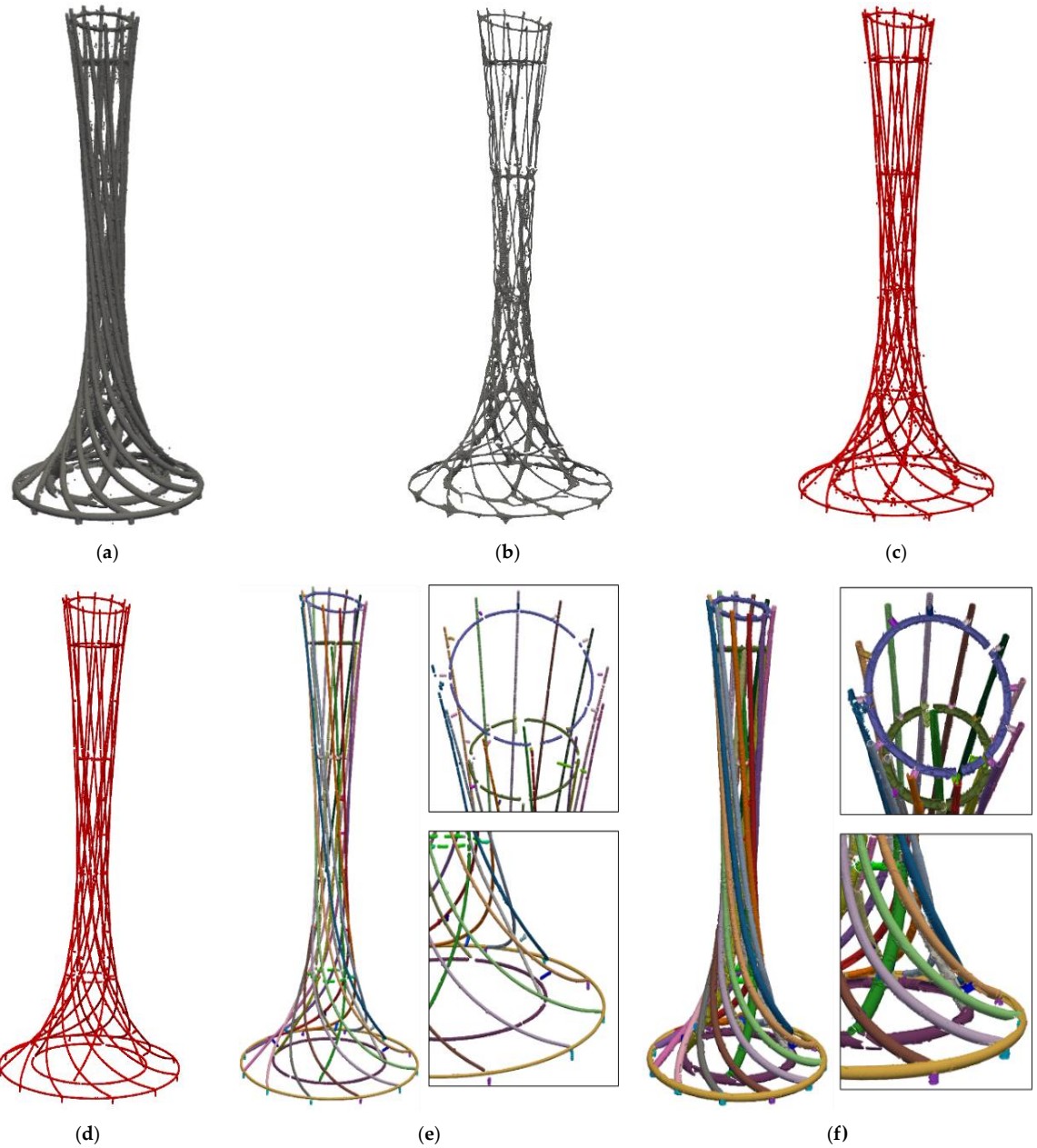

**Figure 15.** Experimental results for the segmentation based on the extracted central axis: (**a**) Input PCD; (**b**) central skeleton contraction; (**c**) candidates on the central axis; (**d**) accurate central axis; (**e**) central axis segmentation result; (**f**) PCD segmentation results.

**Table 2.** The execution time of the 3D model generation procedure.

| Procedure Steps | Number of Points | Execution Time |
| --- | --- | --- |
| Central skeleton contraction | 91,558 | 18,342.58 s |
| Central axis candidate extraction | 91,558 | 5891.21 s |
| Central axis refinement | 91,558 | 948.7 8s |
| Central axis segmentation | 84,733 | 160.12 s |
| PCD segmentation | 21,213,417 | 356.15 s |
| Central axis curve estimation | 84,733 | 2873.21 s |
| 3D modeling of CTES | 21,213,417 | 359.67 s |

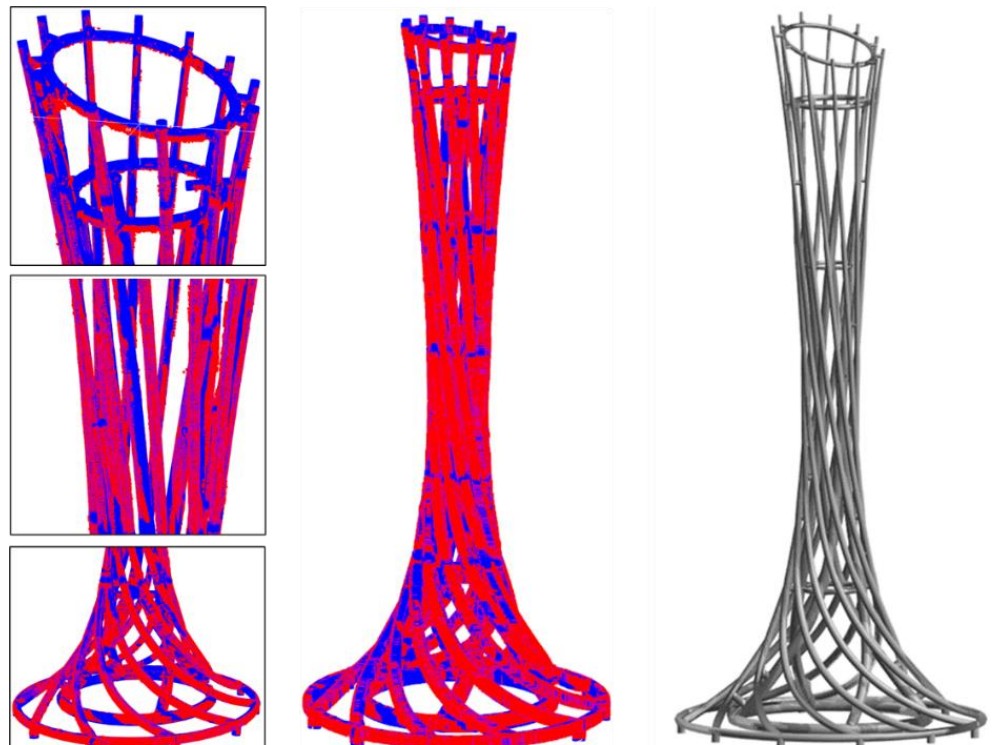

**Figure 16.** The BIM reconstruction of the as-built full-scale CTES.

The sampled PCD with 91,558 points was processed by the central skeleton contraction algorithm to extract the central skeleton, as shown in Figure 15b. Then, the candidates on the central axis were calculated by the rolling sphere algorithm, as shown in Figure 15c. As shown in Figure 15d, the accurate central axis was calculated by the proposed central axis refinement algorithm, where *k* was set to 100 according to [16]. It is worth noting that the number of points on the central axis was 84,733 when the central axis segmentation was performed due to the filtering effect of the central axis refinement algorithm. The results of the central axis segmentation and PCD segmentation are shown in Figure 15e,f, respectively. After the PCD segmentation, 21 curved tubular components and 112 cylindrical joints were obtained. Finally, the proposed geometric parameter estimation and model generation method were used to realize the BIM reconstruction of the as-built CTES. The total running time of the proposed method was about 8 h and 2.2 min, as shown in Table 2. The reconstructed as-built model shown in Figure 16 demonstrates the effectiveness of the proposed method.

### 4.3. Accuracy of Reconstructed BIM

To evaluate the accuracy of the proposed BIM reconstruction method, errors in height and cross-sectional radius of components were calculated. Height error $\delta_h$ and radius error $\delta_r$ were calculated as follows:

$$\delta_h = | \, h_\varepsilon - h_a \, | \tag{11}$$

$$\delta_r = | \, r_\varepsilon - r_a \, | \tag{12}$$

where $h_a$ and $r_a$ are the estimated height and cross-sectional radius of components by the proposed BIM reconstruction method; $h_\varepsilon$ and $r_\varepsilon$ are the measured height and cross-sectional radius of components by Geomagic Wrap [19]. Height errors of 16 curved tubular components are illustrated in Figure 17, where the maximum error is 0.92 mm. Radius errors are given in Figure 18, where the maximum error is 0.21 mm. According to the Chinese code (GB 50205-2020) [42], the allowable error is about 3 mm, demonstrating that the PCD-based results are acceptable in a real application.

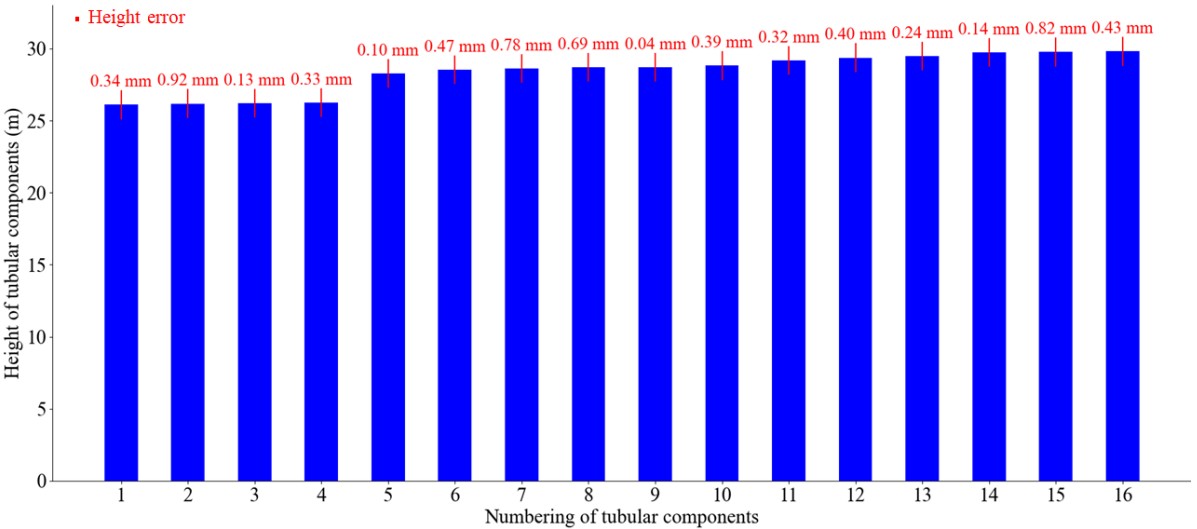

**Figure 17.** Height error.

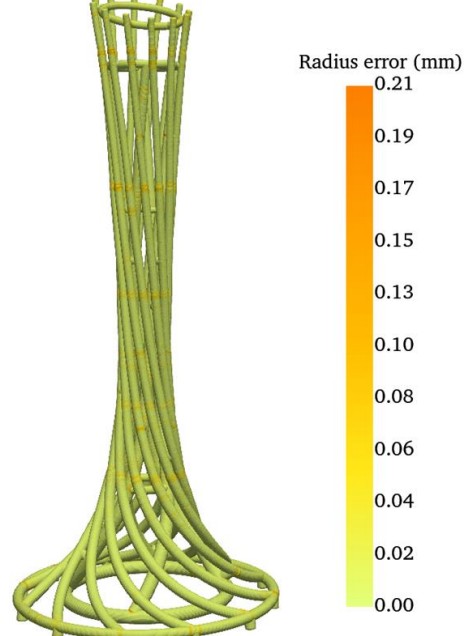

**Figure 18.** Radius error.

### 4.4. Construction Quality Assessment Using the As-Built Model

The as-built model of the CTES obtained can be used for construction quality assessment. Since the shape of the CTES is usually irregular, the construction quality assessment relies on a comparison with the as-designed model [43]. However, it is difficult to find reference points to match the as-designed and as-built models of the CTES. Therefore, this study proposes matching the central axis curve of the as-designed model with that of the as-built model so that the construction quality can be assessed based on the deviation between the nearest neighboring points on the two curves.

To align the as-built model with the as-designed model, the central axis curves of these two models are discretized into points. The centers of circular central axis in these two models are first extracted as shown in Figure 19a,b. Then, coarse alignment is realized based on these extracted centers by using the Procrustes analysis technique [44]. The quality of coarse alignment depends on the accuracy of center estimation. These two curves need to be finely adjusted using the iterative closest point (ICP) algorithm [45] based on all points on the curves, as shown in Figure 19c.

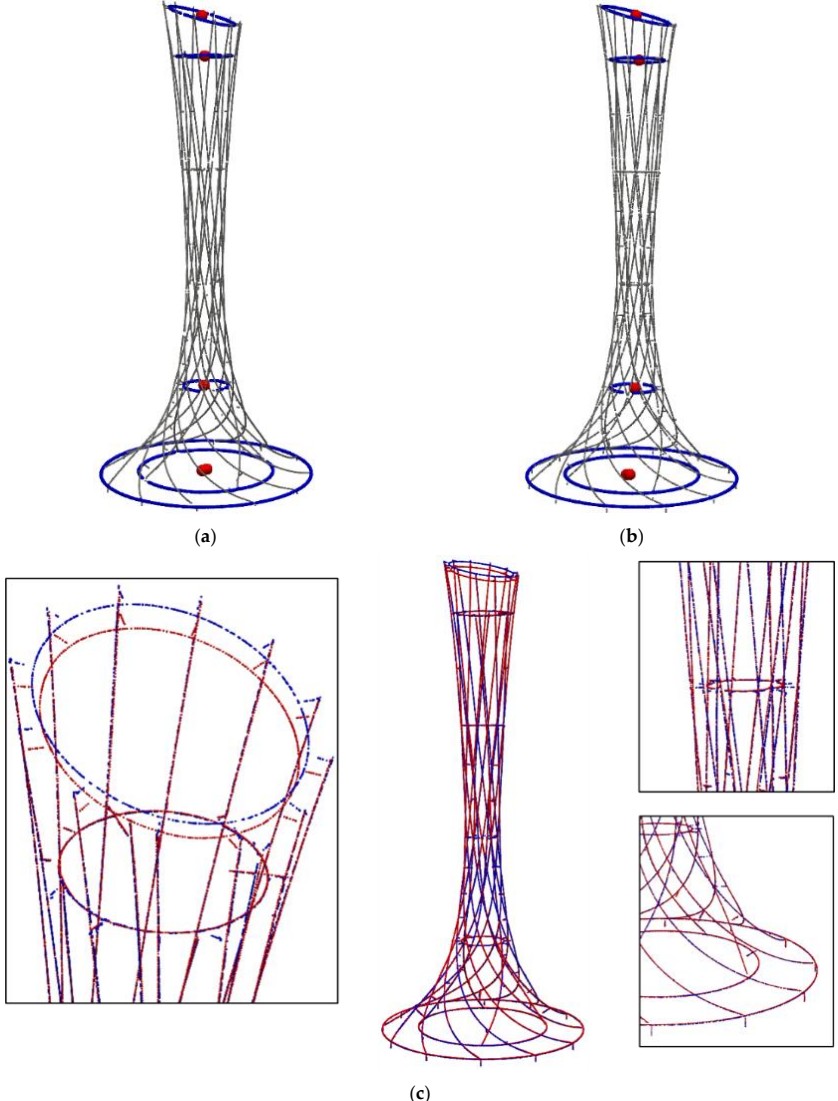

**Figure 19.** The alignment for the central axis curves of the as-built and as-designed models: (**a**) the centers (red) of circular central axis curve (blue) in the as-built model; (**b**) the centers (red) of circular central axis curve (blue) in the as-designed model; (**c**) the alignment result of the as-built model (red) with the as-designed model PCD (blue).

As shown in Figure 19c, there is a significant deviation at the top of the CTES, when the as-designed model matches perfectly with the as-built model. The color-coded deviation map of the CTES is shown in Figure 20, which indicates a maximum deviation at the top of 361 mm. It can be seen that the deviation of the circular tubular component at the top of CETS is relatively even, which is caused by the adjustment of the construction plan. The construction company lowered the circular tubular component by 360 mm, implying that the proposed method can identify deviations in the construction.

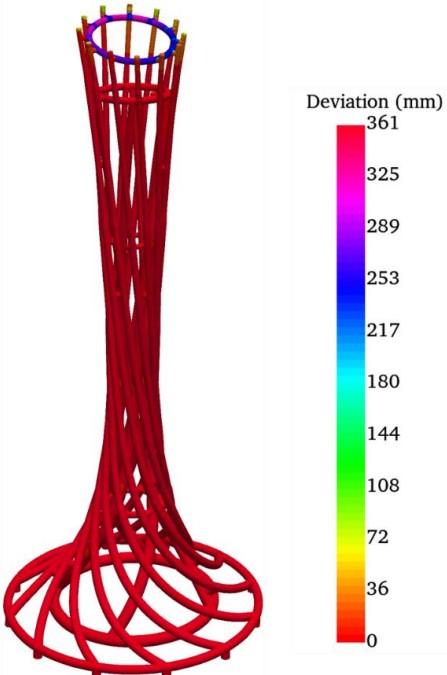

**Figure 20.** The color-coded deviation map of the CTES.

### 4.5. Expanded Experiment

In this section, the above method is verified for a tower structure. The process of BIM reconstruction for this CTES is presented as shown in Figure 21, proving the effectiveness of the proposed method in the BIM reconstruction of other similar CTES.

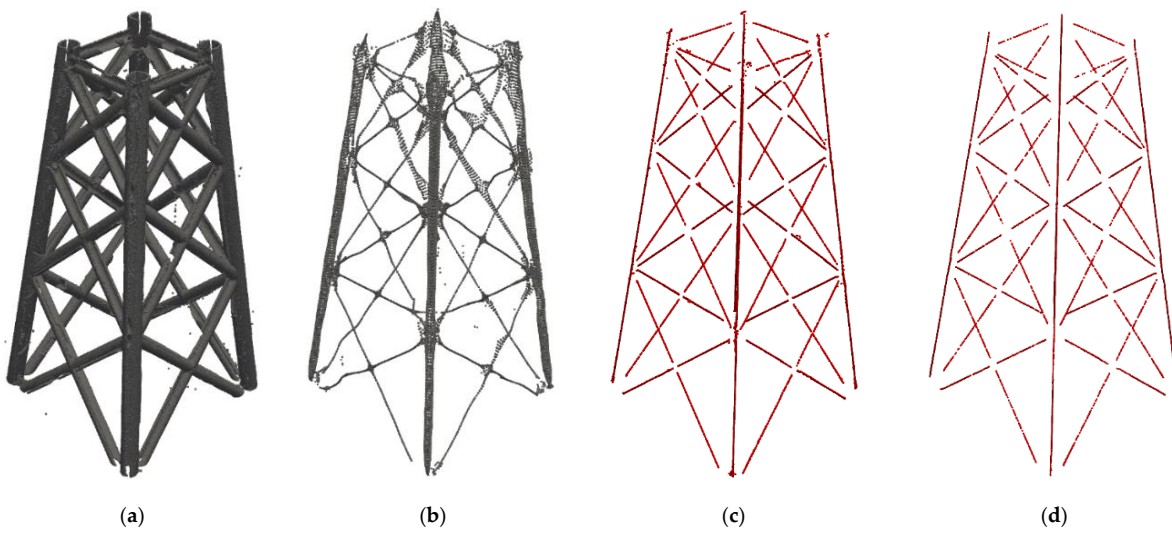

(a)        (b)        (c)        (d)

**Figure 21.** *Cont.*

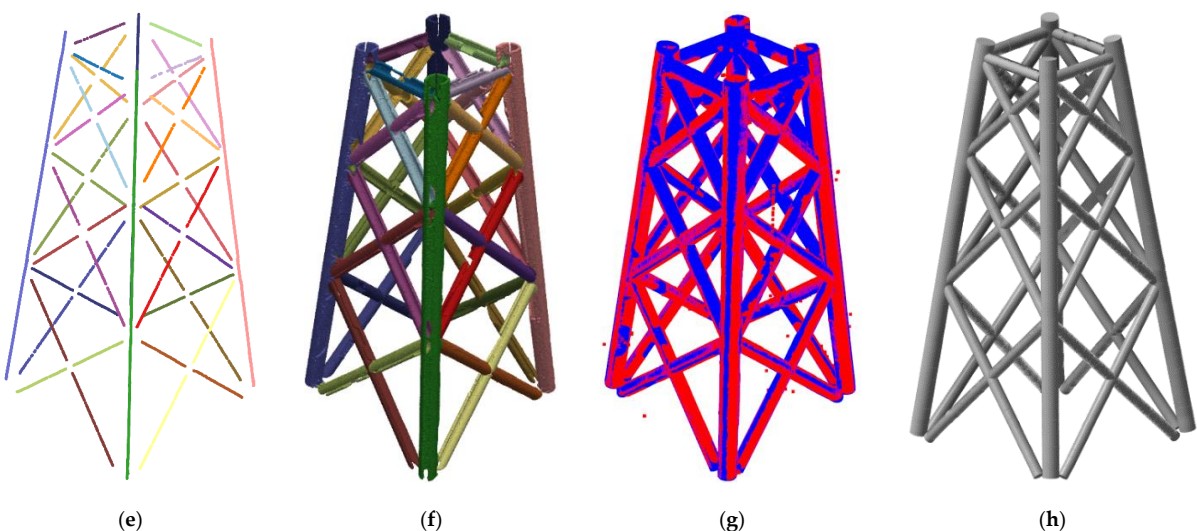

**Figure 21.** The BIM reconstruction of a tower structure: (**a**) input PCD; (**b**) central skeleton contraction; (**c**) candidates on the central axis; (**d**) accurate central axis; (**e**) central axis segmentation results; (**f**) PCD segmentation results; (**g**) geomatic parameter estimation; (**h**) the BIM reconstruction.

## 5. Conclusions

This study proposes an automated BIM reconstruction method for as-built CTESs using terrestrial laser scanning, which includes the central axis extraction, PCD segmentation, geometric parameter estimation, and model generation. The central skeleton of tubular components is extracted first. Based on the points on the central skeleton, the rolling sphere algorithm is used to obtain the candidates on the central axis. A proposed central axis refinement algorithm is further used to obtain an accurate central axis, which can be used for the PCD segmentation. Then, the Catmull–Rom algorithm is adapted for the central axis curve estimation. Next, a slice-based method is used to estimate the cross-sectional radius. Finally, Revit API is adopted to create the as-built model of the CTES in BIM. The validation experiment is conducted on a full-scale CTES. Based on this study, the following main conclusions can be drawn:

(1) A novel central axis extraction method for tubular components is developed and demonstrated to be effective.

(2) An extended axis searching algorithm based on the concept of region growing is proposed, which is suitable for the segmentation of PCD of CETS with missing and noisy data.

(3) The maximum error of the proposed BIM reconstruction method is 0.92 mm, which is acceptable in a real application.

**Author Contributions:** Conceptualization, J.L.; methodology, G.C.; software, N.C.; validation, L.F.; formal analysis, D.L.; investigation, L.F.; resources, J.Z.; data curation, L.F.; writing—original draft preparation, L.F.; writing—review and editing, Y.F.C.; visualization, L.F.; supervision, J.L.; project administration, G.C.; funding acquisition, G.C. All authors have read and agreed to the published version of the manuscript.

**Funding:** The research was funded by the National Natural Science Foundation of China (No.52008055, No.52130801) and the Fundamental Research Funds for the Central Universities (No.2021CDJQY-016). The APC was funded by the National Natural Science Foundation of China (No.52130801). The opinions expressed in this paper belong solely to the authors.

**Data Availability Statement:** Not applicable.

**Acknowledgments:** The authors would like to thank the anonymous reviewers and associate editor for their valuable comments and suggestions to improve the quality of the paper.

**Conflicts of Interest:** The authors declare no potential conflict of interest.

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
