# Peer review of "Automated BIM Reconstruction of Full-Scale Complex Tubular Engineering Structures Using Terrestrial Laser Scanning"

_remotesensing, doi:10.3390/rs14071659_

Round 1
Reviewer 1 Report
The paper addresses a relevant topic of automated BIM reconstruction using laser scanners. In my opinion, this is an interesting and useful topic, since the wider usage of TLS and because the Scan-to-BIM (or the Scan-versus-BIM) is becoming an often-addressed issue.
I found the manuscript well structured and written, and I would suggest just some minor improvements.
Comments:
- the abstract: it would be great to see something more specific about the results from the validation experiment or maybe from the comparison with the as-designed model.
- Figure 1: maybe the text in this figure should have a higher resolution.
- lines 220-222: “To ensure the validity of the fitted sphere, r should be less than 1.5 R and the length of the slice is set to be 0.15R in this study, where R is the maximum cross-sectional radius of the as-designed tubular components in the CETS.”
- why 1.5R and 0.15R? How can you justify it?
- Figure 7: again, the text should have a higher resolution.
- Algorithm 1 – what is the value of the angle threshold α? Based on what it is determined?
- lines 289-290: “In this study, ω is set to be 0.05 and ε is set to be 0.1”
- How can you explain the choice of these values?
- lines 307-308: “Where the parameter τ is known as “tension” which affects how sharply the curve bends at the 4 points, set as 0.2 in this study.”
- again, why was the value of parameter τ chosen as 0.2? How can you justify it?
- lines 338-340: it would be great to see some basic parameters from the measurement with the mentioned scanner, e.g., the resolution, accuracy, etc.
- line 384: “… where the number of neighboring points is set as 100”
- why 100? Maybe some explanation would be fine.
Reviewer 2 Report
This study develops a method to achieve automated BIM reconstruction for full-scale complex engineering structures. The designed algorithm is improved from previous studies and is tested on a practical case. The overall research is interesting. However, there are still some questions that need to be addressed.
- Line 53: Why are the central axis and cross-sectional radius critical?
- Line 63: Why did only a few studies conduct on the BIM reconstruction of full-scale CTEs? Besides, although previous studies may not consider BIM reconstruction of CTEs, their methods can be applied for BIM reconstruction of CTEs. Please explain.
- Line 71: The 3rd point is not a contribution.
- Line 153 – line 154: Why the central skeleton contraction algorithm and rolling sphere algorithm are adopted to solve the problems in this study?
- What are the differences of CTEs compared to the other elements in previous studies? Other than the deficiency of the previous methods, did the algorithms designed in this study consider the specific features of CTEs?
- What is the tolerance of the PCD-based result? How do you ensure that PCD-based result is acceptable in a real application?
- Please present the error (e.g., length) of the extraction results.
- The calculation result (i.e., the maximum deviation is 361mm) should be validated by other methods or experts to demonstrate that this result is reasonable, and the proposed method is effective.
- It seems that the proposed method is designed specifically for the validation case. Please provide more experiments or explanation that this method can be applied to other similar structures.
- Line 442: This should not be a conclusion of this study but a result of applying the proposed method. The authors should indicate that the proposed method is able to identify the deviations.
Reviewer 3 Report
This study presents curve fitting methods for automated BIM reconstruction for CETSs. In overall, the algorithms are appropriately explained and the results are excellent. The results are supported with enough data, so the proposed methods are practical to be applied to real structures. Based on the soundness and feasibility of this study, the reviewer has no objection for the publication of this paper as it is.
Author Response
Dear reviewer,
Thanks very much for taking your time to review this manuscript. I really appreciate all your comments and appreciation for this manuscript!
Thanks again!

Reviewer 4 Report
The authors provide an improved method for comparing as-built vs. as-designed models based on extended-axis searching algorithm in point cloud data (PCD) for complex tubular engineering structure (CTES). The paper significantly improves the currently used methods based on the improvement of accuracy for this purpose, especially in the case of missing and noisy PCD. However, the authors are advised to supplement the article with minor corrections and amendments in the sections as set out below.
Lines 339 to 340 - the real-world CETS was scanned using FARO S150 and 5 scans were captured at different locations. It is not explained why exactly 5 scans were performed and it is not stated which are the different locations of the scan or. how they were determined. The number of scans and pre-optimized capture location can significantly affect PCD results. It is further stated that all scans were registered and preprocessed using a specialized software. There is no precise data on the registration process, which algorithms were used, and what is the result of the registration, ie. what is the accuracy of the registration. There is also no indication of what the point cloud quality requirement is for further evaluation of the proposed method.
Round 2
Reviewer 2 Report
This version of the manuscript has clearly addressed the comments in the last round.